# Mechanisms, Characteristics, and Treatment of Neuropathic Pain and Peripheral Neuropathy Associated with Dinutuximab in Neuroblastoma Patients

**DOI:** 10.3390/ijms222312648

**Published:** 2021-11-23

**Authors:** Stefano Mastrangelo, Serena Rivetti, Silvia Triarico, Alberto Romano, Giorgio Attinà, Palma Maurizi, Antonio Ruggiero

**Affiliations:** Pediatric Oncology Unit, Fondazione Policlinico Universitario Agostino Gemelli IRCCS, Università Cattolica Sacro Cuore, 00168 Rome, Italy; serena.rivetti@gmail.com (S.R.); silviatriarico@libero.it (S.T.); albertoromano90.ar@gmail.com (A.R.); giorgio.attina@policlinicogemelli.it (G.A.); palma.maurizi@unicatt.it (P.M.); antonio.ruggiero@unicatt.it (A.R.)

**Keywords:** dinutuximab, neuropathic pain, peripheral neuropathy, neuroblastoma, pediatric cancer, molecular mechanisms, treatment

## Abstract

Prognosis of metastatic neuroblastoma is very poor. Its treatment includes induction chemotherapy, surgery, high-dose chemotherapy, radiotherapy, and maintenance with retinoic acid, associated with the anti-GD2 monoclonal antibody (ch14.18) dinutuximab. Immunotherapy determined a significant improvement in survival rate and is also utilized in relapsed and resistant neuroblastoma patients. Five courses of dinutuximab 100 mg/m^2^ are usually administered as a 10-day continuous infusion or over 5 consecutive days every 5 weeks. Dinutuximab targets the disialoganglioside GD2, which is highly expressed on neuroblastoma cells and minimally present on the surface of normal human neurons, peripheral pain fibers, and skin melanocytes. Anti GD2 antibodies bind to surface GD2 and determine the lysis of neuroblastoma cells induced by immune response via the antibody-dependent cellular cytotoxicity and the complement-dependent cytotoxicity. Dinutuximab has significant side effects, including neuropathic pain, peripheral neuropathy, hypersensitivity reactions, capillary leak syndrome, photophobia, and hypotension. The most important side effect is neuropathic pain, which is triggered by the same antibody–antigen immune response, but generates ectopic activity in axons, which results in hyperalgesia and spontaneous pain. Pain can be severe especially in the first courses of dinutuximab infusion, and requires the administration of gabapentin and continuous morphine infusion. This paper will focus on the incidence, mechanisms, characteristics, and treatment of neuropathic pain and peripheral neuropathy due to dinutuximab administration in neuroblastoma patients.

## 1. Introduction

Neuroblastoma is the most common extracranial solid tumor of childhood, with a median age at diagnosis of 17 months. Its incidence is 10.2 cases per million children aged <15 years. Neuroblastoma arises in tissues of the sympathetic nervous system, mostly in the adrenal medulla or paraspinal ganglia. It appears as a mass in the abdomen, pelvis, neck, or chest, with about half of the patients having metastatic disease at diagnosis [1]. The presence of metastatic diseases over the age of 12 or 18 months and aggressive biological features (e.g., *MYCN* oncogene amplification) define high-risk neuroblastoma [2,3]. The prognosis for such patients is poor, with a long-term survival rate of only 40% [4].

The treatment strategy for high-risk neuroblastoma patients includes induction chemotherapy, surgery, consolidation with myeloablative high-dose chemotherapy (HDT) followed by autologous stem cell transplantation (SCT), and maintenance therapy. Better tumor responses after induction therapy appear to be critical to improve the percentage of long-term survival, but the dose intensity of traditional drugs such as platin compounds, cyclophosphamide, etoposide, doxorubicin, vincristine, topotecan, and temozolomide cannot be increased because of their hematological and non-hematological toxicities [5,6,7,8,9,10,11,12,13]. Moreover, some new combinations of agents active against neuroblastoma have been developed with encouraging results, but none of them have yet been adopted in large randomized trials [14,15,16].

One of the main causes of treatment failure is the presence of minimal residual disease after the end of first line treatment. Targeted immunotherapy given at maintenance has been shown to be efficacious in removing residual disease and, therefore, to improve clinical outcome [1].

Dinutuximab (ch14.18) is a human-murine chimeric antibody formed by a variable region from murine anti-GD2 antibody 14G2 fused with a constant region from human IgG1 antibody. It is generated in murine myeloma cells SP2/0 and contains murine retroviruses. Dinutuximab targets the disialoganglioside GD2, which is highly expressed on neuroblastoma cells, contributing to the binding of tumor cells to the extracellular matrix. Moreover, GD2 is only minimally present on the surface of normal human neurons, peripheral pain fibers, and skin melanocytes, and was thus considered an attractive target for anti-GD2 immunotherapy [17].

Several studies have demonstrated that the addition of anti-GD2 antibody as immunotherapy in the maintenance phase improves survival in patients affected by high-risk neuroblastoma. The most important study was published by Yu et al. in 2010; it reports on the Children’s Oncology Group (COG) ANBL0032 phase 3 trial conducted on 226 high-risk neuroblastoma patients treated at diagnosis. Patients that responded to induction treatment and HDT were randomized to receive isotretinoin alone or treatment with anti-GD2 antibody ch14.18 plus interleukin-2 (IL-2), granulocyte-macrophage colony-stimulating factor (GM-CSF), and isotretinoin. Immunotherapy determined a 20% increase in event-free survival (EFS) at 2 years and an 11% increase in overall survival (OS) at 2 years compared with isotretinoin alone [18].

The International Society of Pediatric Oncology Europe Neuroblastoma (SIOPEN) group ordered to produce dinutuximab in Chinese hamster ovary (CHO) cells; the ch14.18/CHO was named dinutuximab beta (Qarziba^®^, Schiphol-Rijk, The Netherlands). This chimeric antibody presented a more favorable glycosylation pattern to avoid the clearance by xeno-autoantibodies and murine xenotropic retrovirus contamination. Dinutuximab beta was then approved for treatment by the European Medicines Agency [19].

The SIOPEN group started a trial to explore the effects of immunotherapy with dinutuximab beta in high-risk neuroblastoma (HR-NBL1 protocol). The first patients enrolled from 2006 were randomized to receive isotretinoin with dinutuximab beta or isotretinoin alone. However, the study was stopped after the first results of COG ANBL0032 trial were reported, showing the superiority of treatment with dinutuximab and isotretinoin over isotretinoin alone. Therefore, from 2009, in the same HR-NBL1 protocol, a new randomization was opened to investigate if the addition of IL-2 to patients treated with dinutuximab beta and isotretinoin improved outcome.

The results reported by Ladenstein et al. showed superior five-year EFS and OS when dinutuximab beta-based immunotherapy with or without IL-2 was included in maintenance therapy, compared with the group treated with isotretinoin alone. The EFS was 57% for the first group compared with 42% for the second one, while the five-year OS for the first group was 64% compared with 50% for the second group [20].

The role of IL-2 administered in association with dinutuximab beta is controversial, even though the results of the new randomization in SIOPEN trial, regarding the addition of IL-2 to dinutuximab beta and isotretinoin, showed that IL-2 did not improve outcome, but increased toxicity. These findings were reported in a paper by Ladenstein et al. [21], in which authors concluded that dinutuximab beta and isotretinoin without subcutaneous IL-2 is to be considered the standard care for high-risk neuroblastoma patients after induction and HDT with SCT.

Dinutuximab beta is approved in the European Union for the treatment of high-risk neuroblastoma in pediatric patients aged >12 months. It can be used both in patients who have achieved at least a partial response to the induction chemotherapy followed by myeloablative therapy and SCT and in patients who have relapsed or refractory neuroblastoma with or without residual disease, after stabilization of any actively progressing disease.

Dinutuximab beta is administered at a total dose of 100 mg/m^2^ per course, for five consecutive courses of 35 days each. It is given at a daily dose of 10 mg/m^2^ as a continuous infusion over the first 10 days of each course or in infusions of 20 mg/m^2^/day over 8 h, on the first 5 days of each course.

For patients with relapsed or refractory neuroblastoma, IL-2 should be added to dinutuximab beta, by subcutaneous injections of 6 × 10^6^ IU/m^2^/day, for two periods of 5 consecutive days. The first 5 days of treatment start 7 days before the first infusion of dinutuximab beta, while the other 5-day course is administered concurrently with dinutuximab beta infusion [22].

All patients should also receive isotretinoin for 14 days starting one day after the end of each dinutuximab beta course.

Treatment with dinutuximab beta has significant side effects, such as neuropathic pain, peripheral neuropathy, hypersensitivity reactions, capillary leak syndrome, photophobia, and hypotension. The risk of developing these symptoms increases when administration of IL-2 is associated with dinutuximab beta in comparison with dinutuximab beta given alone [21].

Pain is mediated by the reaction of dinutuximab beta with the GD2 antigen on the surface peripheral nerve fibers [23]. In children, pain is less severe and less opiate resistant; this may be because of the relatively greater GD2 enrichment of cellular membranes following maturation [24].

In this review, we analyze the use of dinutuximab in the treatment of high-risk neuroblastoma, with a focus on the mechanisms and strategies of prevention and treatment of dinutuximab-associated neuropathic pain and peripheral neuropathy in pediatric patients. We have searched for papers dedicated to his topic in the pediatric age, performing a Pubmed-based retrieval of articles using the search terms “dinutuximab”, “immunotherapy”, “high-risk neuroblastoma”, “toxicity”, “transient peripheral neuropathy”, and “pain” matched with “children”, “childhood”, and “pediatric”. After the original search, we used filters to select articles available in the English language and articles with available full texts.

## 2. GD2 Function and Anti-GD2 Antibodies

Dinutuximab beta is a chimeric monoclonal IgG1 antibody targeting the carbohydrate moiety of GD2, which derives from ceramides and is highly expressed on neuroblastoma cells (Figure 1).

Gangliosides have an important role in signal transduction as well as cell adhesion and recognition [25]. They are distinguished in a-series, generally expressed in normal tissue, and b-series. In normal tissue, b-series gangliosides are normally expressed during fetal development on neural and mesenchymal stem cells; in adults, they are restricted primarily to the nervous system and at low levels in peripheral nerves and skin melanocytes [26]. GD2 is a b-series ganglioside highly expressed on neuroectoderm-derived tumors (neuroblastoma, retinoblastoma, melanoma, small cell lung cancer, and brain tumors) and sarcomas (rhabdomyosarcoma, osteosarcoma, and Ewing’s sarcoma).

In normal tissue, GD2 expression is restricted to the brain, which is inaccessible to circulating antibodies, as well as to peripheral nerve fibers and melanocytes. It is expressed on the outer cell membrane and seems to play a role in neuronal development, differentiation, and repair [27]. In tumor tissue, it appears to be involved in proliferation, invasion, angiogenesis, metastasis, and immunity, as observed in small cell lung cancer cells and osteosarcoma cells [28,29]. Cazet et al. showed that up-regulation of GD2 in triple negative (ER-, PR-, and Her-) breast cancer cells led to enhanced proliferation of the tumor [30]. It has been suggested that GD2 enhances the proliferative ability of GD2+ tumor cells by inducing constitutive activation of the proto-oncogene c-Met.

GD2 seems to contribute to the binding of tumor cells to the extracellular matrix [17]. Another mechanism suggests that GD2 enhances platelet adhesion to extracellular matrix collagen by upregulating integrin α2β1-mediated tyrosine phosphorylation of p125FAK; through this way, it seems to promote metastasis of neuroblastoma cells [31].

As GD2 has been detected as a target for neuroblastoma, several types of antibodies were developed. The most promising ones, used in clinical trials, were a murine anti-GD2 antibody (3F8), a chimeric anti-GD2 antibody (ch14.18), and a humanized anti-GD2 antibody (hu14.18K322A) [32]. At first, in phase I and II studies, promising antitumor effects were obtained testing the murine anti-GD2 antibody 3F8 either alone or with GM-CSF in patients with melanoma or neuroblastoma [33,34]. The best apparent outcome resulted from combining 3F8 with subcutaneous GM-CSF and isotretinoin [35]. Moreover, the combination of another murine anti-GD2 antibody (ch14.G2a) with intravenous IL-2 showed a partial antitumor response in one neuroblastoma patient (out of 33 patients) and a decrease in neuroblastoma cells in the bone marrow of three patients [36].

Subsequently, humanization of anti-GD2 antibodies was undertaken to diminish the development of human anti-mouse antibodies, potential blockers of antitumor activity of anti-GD2 antibodies. A partially humanized antibody was developed, that is, the chimeric ch14.18, proved to be better tolerable than the murine antibody [37]. Subsequently, ch14.18 antibody produced in CHO cells (ch14.18/CHO) called dinutuximab beta (Qarziba^®^), was produced in Europe [19]. Finally a humanized anti-GD2 antibody (hu14.18K322A), engineered with a point mutation (lysine to alanine) at position 322 in the Fc domain of the antibody, was designed to reduce the complement activation, and thus to induce less pain [32].

## 3. Pharmacodynamic and Pharmacokinetic of Anti-GD2 Antibodies

Binding of anti-GD2 monoclonal antibodies to tumor cells interferes with proliferation and invasiveness of the tumor, and directly induces apoptosis [38]. The abundant expression of GD2 on neuroblastoma, but limited expression on normal cells, made it an appealing target for anti-GD2 immunotherapy.

The pharmacokinetic characteristics of dinutuximab beta were evaluated in phase 1 and phase 2 studies. Maximum plasma concentrations for 8 h infusion regimen (20 mg/m^2^/day over 5 consecutive days) were reached on the fifth day of the infusion. Maximum plasma concentrations during a long-term infusion (continuous infusion of 10 mg/m^2^ over 10 days) were reached on the last day of the infusion [22]. Values for the area under the serum concentration–time curve and clearance with the long-term infusion were not significantly different from those observed with the short-term infusion [39].

Dinutuximab beta seems to be degraded by ubiquitous proteolytic enzymes to small peptides and individual amino acids. The half-life of dinutuximab beta observed in phase 1 and 2 studies is 8 days [22].

The anti-cancer activity of dinutuximab beta takes place through interaction with components of the patient’s immune response mechanisms. Dinutuximab beta binds to surface GD2 and triggers immune mechanisms via the antibody-dependent cellular cytotoxicity (ADCC) and the complement-dependent cytotoxicity (CDC)., leading to the lysis of neuroblastoma cells [17] (Figure 2g–i).

In fact, the majority of patients show a progressive complement (C4 and C3) depletion over the course of therapy due to its consumption [24].

Concerning the first mechanism, Natural Killer (NK) cells can kill antibody-bound neuroblastoma cells. The addition of cytokines like IL-2 to activate NK cells or other immune cells in the tumor microenvironment is one way to stimulate ADCC. The addiction of GM-CSF to increase the activation of myeloid cells is another mechanism to enhance ADCC against neuroblastoma cells [35]. Macrophages and neutrophils are other effector cells of ADCC. Macrophages are also able to phagocytize neuroblastoma cells and participate in the antitumor response [40].

A third cell type involved in ADCC is γδ T cells; they induce neuroblastoma cell death in vitro and in vivo in mouse studies [41].

## 4. Dinutuximab Induced Neuropathic Pain

Neuropathic pain affects from 33% to 88% of pediatric patients undergoing dinutuximab infusion. Pain usually manifests as allodynia, which involves various regions of the body, especially the abdomen, extremities, back, and chest, and then spreads peripherally to the ankles and feet [18,32,37,42,43,44]. Anti-GD2 induced pain is characterized by mechanical allodynia without thermal hyperalgesia [23].

It usually begins within an hour from the start of dinutuximab beta infusion and is limited to the time of administration of this drug, ending shortly after the termination of the infusion; it usually occurs during the first infusion of the drug and decreases after each course [22]. A retrospective study on 26 patients affected by high-risk neuroblastoma who received dinutuximab based immunotherapy concluded that grade ≥3 (of a 1 to 5 scale) pain occurs in about 88% of patients during immunotherapy course 1, but in 42% of patients during course 5 [42].

GD2-induced neuropathic pain is mediated by the reactivity of the antibody with the GD2 antigen on the surface of peripheral nerve fibers, particularly C-fibers [23]. The inability of GD2 antibodies to cross the blood brain barrier implicates the involvement of the peripheral nervous system [23].

The pain-inducing mechanism is unclear. It probably involves the same immune response by ADCC and CDC elicited for treatment; the antigen–anti-GD2-antibody complex on the GD2-expressing nerve fibers is thought to be the first trigger for pain (Figure 2a–c).

Antibodies and immune cell products from the antibody–antigen response generate ectopic activity in axons, which results in hyperalgesia and spontaneous pain [24].

It is possible that patients who benefit from dinutuximab treatment are those with a high percentage of neuroblastoma cells that are GD2 positive [45]. Nevertheless, not all patients with progressive neuroblastoma will have high expression of GD2 on tumor cells [46]. Thus, such patients may have toxicity with no benefit from dinutuximab. Furthermore, the selective pressure of dinutuximab therapy may result in decreased GD2 expression, which has already been observed with targeting CD20 on lymphoma with rituximab, targeting CD19 on leukemia with CAR-T cells, and with antibodies to EGF in breast cancer [47,48,49].

### Prevention and Treatment

Because of the high frequency of neuropathic pain during dinutuximab beta infusion, various strategies of prevention and treatment have been evaluated.

First, a continuous 10-day infusion appeared more tolerable than the discontinuous 5-day infusions on consecutive days. Mueller et al. compared data from 53 patients with high-risk neuroblastoma who received continuous 10-day infusions of dinutuximab beta with those of 226 patients from the study by Yu et al. who received discontinuous once-daily infusions of dinutuximab on 4 consecutive days [18,50]. The incidence of treatment-related adverse events was lower with the continuous infusion than with the once-daily infusion; particularly, the incidence of neuropathic pain decreased from 51.8% of patients under once-daily infusion to 37.7% of patients under continuous infusion.

Furthermore, as one of the mechanisms of generation of neuropathic pain seems to be the complement activation at the GD2-expressing nerve fibers level [51], humanized anti-GD2 antibodies (hu14.18K322A) have been shown to resolve pain more rapidly, through reduction of complement activation, even though opioid requirements were not reduced [32,52]. Moreover, initial data show some complete responses in the treatment of recurrent or refractory neuroblastoma, but a randomized trial is needed to determine if the elimination of complement binding may maintain anti-GD2 activity in addition to decreasing neuropathic pain [53,54].

Aggressive pain control is such a priority that the U.S. Food and Drug Administration recommends permanent discontinuation of dinutuximab in patients with severe pain that is uncontrollable by analgesic therapy.

Because neuropathic pain usually occurs at the beginning of the treatment, its onset can be predicted and premedication with analgesics should be administered. Triple therapy with gabapentin, non-opioid analgesics, and opioids is recommended for pain treatment [22] (Figure 3). Three days prior to dinutuximab beta infusion, oral gabapentin administration should be started at the dose of 10 mg/kg/day. The next day, the dose is increased to 10 mg/kg for two daily administrations and the next day to 10 mg/kg for three daily administrations (the maximum single dose of gabapentin must not exceed 300 mg). Gabapentin reverses the tactile allodynia that follows anti-GD2 administration; its administration should be continued as long as required by the patient and be tapered off up to suspension after weaning off intravenous morphine infusion [24].

Non-opioid analgesics such as paracetamol or ibuprofen should be used during the treatment.

The use of opioids for the duration of antibody therapy given as continuous infusion over 10 days is essential for pain control, with higher opioid doses given in the first infusion day and course than in subsequent days and courses. Morphine should be commenced 2 h before dinutuximab beta infusion with a bolus of 0.02–0.05 mg/kg. Subsequently, continuous intravenous morphine infusion should be started at a dosing rate of 0.03 mg/kg/h and continued during all dinutuximab beta infusion. In response to patient’s pain perception, it may be necessary to increase the infusion rate or, on the other hand, it could be possible to wean off morphine over 5 days, progressively decreasing its dosing rate. If continuous morphine infusion is required for more than 5 days, treatment should be gradually reduced by 20% per day after the last day of dinutuximab beta infusion.

In the case of neuropathic pain persisting after the intravenous morphine is weaned off, oral morphine sulphate (0.2–0.4 mg/kg every 4–6 h) or tramadol (for moderate neuro-pathic pain) can be administered on demand.

For subsequent courses, analgesic therapy at the highest dose of opioid required for adequate pain control during the prior course should be considered at the start and then modulated according to the severity of pain [55].

Other analgesic regimens were experimented but, to our knowledge, the previous is the best strategy for prevention and treatment of dinutuximab beta-related neuropathic pain. Opioid transdermal delivery system has not been utilized as it is not adjustable on the basis of patients’ variable pain intensity [56,57].

The concurrent administration of low-dose lidocaine has been shown to reduce opioid consumption in neuroblastoma patients treated with immunotherapy, but it determined a very high incidence of vomiting and may not be appropriate in the ward setting [24]. Gorges and colleagues studied the use of dexmedetomidine and hydromorphone to manage the pain associated with anti-GD2 infusion. They reported adverse effects such as hypotension, hypoxemia, and bradycardia in 30%, 8%, and 4% of treatment days, respectively [58]. Bertolizio et al. retrospectively analyzed the efficacy of a multimodal regimen with gabapentin, ketamine, and morphine for preventing and treating neuropathic pain during dinutuximab therapy [59]. Even if the addition of ketamine to opioids is known to decrease morphine consumption and pain, this study showed higher morphine consumption and a lower incidence of moderate and severe pain with fewer adverse effects in comparison with data by Georges et al.; this may be due to avoidance of dexmedetomidine [58,60].

## 5. Dinutuximab Induced Peripheral Neuropathy

Peripheral neuropathy of any kind results from a partial loss of innervation and it can also lead to chronic pain. Neuropathy is due to a physical, metabolic, or chemical injury of axons that determines a disconnection of nerve branches from their terminals.

In a normal pathway, these injuries lead to Wallerian degeneration and apoptosis. Then, dedifferentiation of resident Schwann cells and the infiltration of systemic immune cells open the way for axonal regeneration and nerve repair. If the balance between degeneration and regeneration is interrupted, chronic pain may begin [61,62] (Figure 2d–f).

Peripheral neuropathy is a common side effect of chemotherapeutic agents, such as oxaliplatin and vincristine, through different mechanisms. Oxaliplatin accumulates in dorsal root ganglion neurons and causes functional changes in nerve excitability to undergo severe neuropathy [63]. Vincristine determines a compromission of microtubule function and axonal transport with mitochondrial toxicity [7,64]. Another pathway observed in the chemotherapeutic drugs neurotoxicity is the stimulation of NK cells by upregulation of stress related proteins in tumor and/or downregulation of inhibitory self-ligands on the target cell surface [65,66,67].

Peripheral neuropathy is a rare side effect of dinutuximab beta administration; it can be severe with a prevalence of between 2 and 6% of patients [42]. Ladenstein et al. reported seven patients with grade 3 and 4 toxicities, the most frequent being paresthesia or deficits in motor function; one patients with tetraparesis improved over time, but did not recover completely [21]. Mody et al. described one patient treated with irinotecan, temozolomide, and dinutuximab that presented with grade 3 peripheral motor neuropathy started on day 6 of cycle 6. He suffered a bilateral lower extremity weakness and was unable to walk independently for 4 weeks, but completely recovered after 2 weeks [68]. One patient with grade 3 peripheral motoneuropathy was reported by Mueller et al., while another patient with peripheral neurotoxicity, which did not resolve completely, was seen by Blom et al. [42,50].

The mechanisms of anti-GD2 neurotoxicity remain unclear. It is certain that anti-GD2 antibodies bind the GD2 antigen localized on the cell membrane of axons of peripheral nerves and affect its function in vivo [69,70]. Subsequently, activated by anti-GD2 antibodies via ADCC, NK cells seem to have a potential pathogenic role by forming an immunological synapse with the axon of a sensory neuron and thus releasing ions and proteases into the target neuron; this would lead to axon microtubule destabilization and axon degeneration [71].

A sensorimotor polyneuropathy associated with demyelination and a mononuclear infiltrate in both endoneurial and perivascular space of sural nerve was documented in adults after treatment of metastatic melanomas with an anti-GD2 monoclonal antibody [72].

A study of Yuki et al. tried to clarify what causes the neurotoxicity of anti-GD2 monoclonal antibody. They showed that anti-GD2 reacts with the peripheral nerve myelin and assumed that this binding causes antibody-dependent cell mediated and complement-dependent demyelination, resulting in the development of sensorimotor demyelinating polyneuropathy [72].

As anti-GD2 modified to reduce complement activation determined the reduction but not the elimination of side effects on peripheral nervous system in a rat model, other mechanisms are probably involved, such as an immune-mediated inflammation [27,51,53,54]. However, the complete mechanism needs to be clarified.

### Treatment

If dinutuximab beta infusion determines motor or sensory peripheral neuropathy that lasts for more than 4 days, a non-inflammatory cause should be considered and excluded, as it could be expression of neuroblastoma progression, infection, metabolic syndrome, and concomitant medication.

Exams like magnetic resonance and motor and sensory electromyography studies should be conducted.

To the best of our knowledge, there are no studies that specifically address the treatment of dinutuximab induced peripheral neuropathy. In patients with grade 2 neuropathy, dinutuximab beta infusion should be interrupted, to be resumed only after neurologic symptoms resolve. Infusion must be discontinued if subjects present any objective prolonged weakness possibly as a result of dinutuximab beta administration [22,73]. If symptoms do not improve after a few days, steroid therapy should be considered.

## 6. Conclusions

Immunotherapy with anti-GD2 antibody dinutuximab has become an essential component of treatment for high-risk neuroblastoma patients, even though its side effects are very frequent and often severe. The most important adverse reaction is neuropathic pain, which, as for peripheral neuropathy, is triggered by an immune reaction of dinutuximab with normal peripheral neurons that express GD2 on their surface membrane. Various strategies for prevention and treatment have been evaluated to control pain, the best one seems to be multimodal therapy that includes continuous infusion of morphine associated with gabapentin and non-opioid analgesics administered concurrently with dinutuximab continuous infusion over 10 days. With the introduction of dinutuximab beta and humanized anti-GD2 antibodies, there has been some progress in the development of a more tolerable but yet efficacious immunotherapy. It is hoped that new immunological agents active against high-risk neuroblastoma will be developed in the near future, so as to have more powerful weapons available against a tumor that still carries a very bad prognosis.

## Figures and Tables

**Figure 1 ijms-22-12648-f001:**
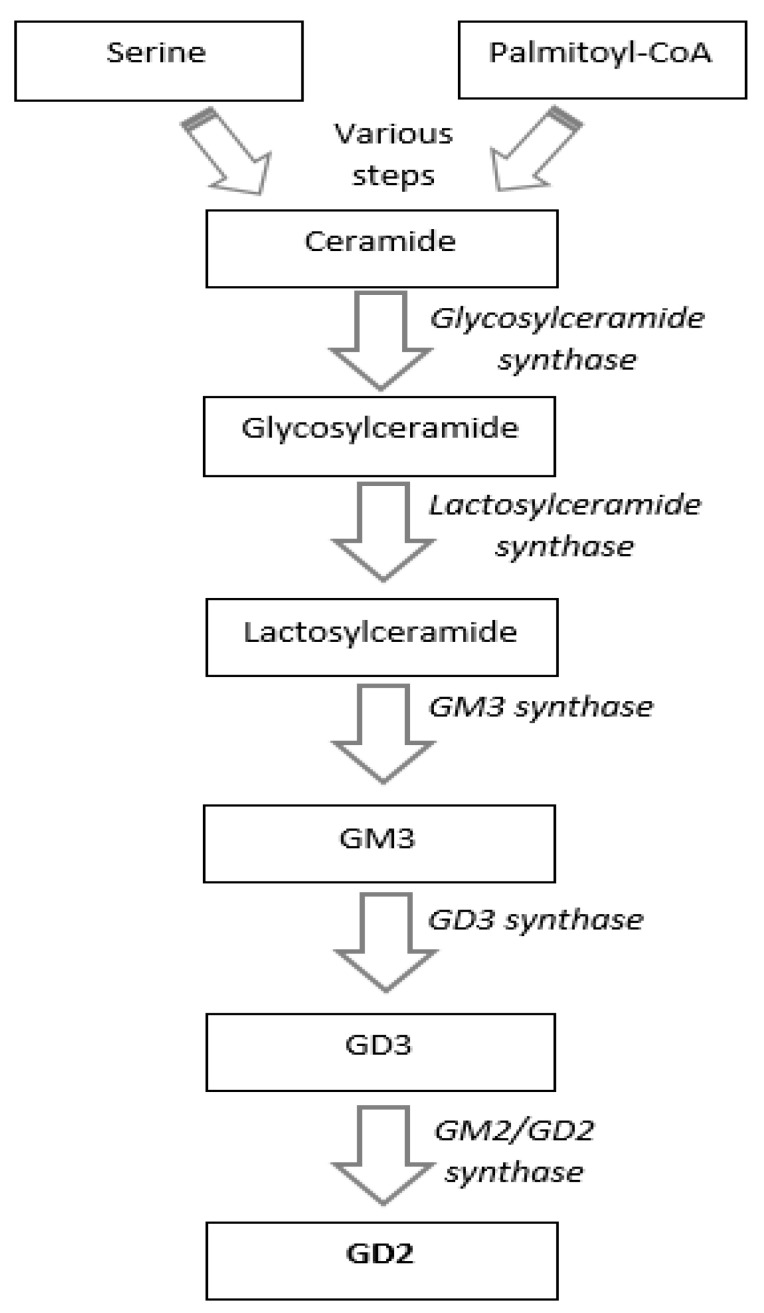
GD2 is a disialoganglioside obtained from ceramides via glycosylations. Ceramides are synthetized from serine and palmitoyl-CoA after various steps. The first glycosylation produces glycosylceramide, which is glycosylated in lactosylceramide. This is converted to GM3 by GM3 synthase, GM3 to GD3, and GD3 to GD2 by GM2/GD2 synthase.

**Figure 2 ijms-22-12648-f002:**
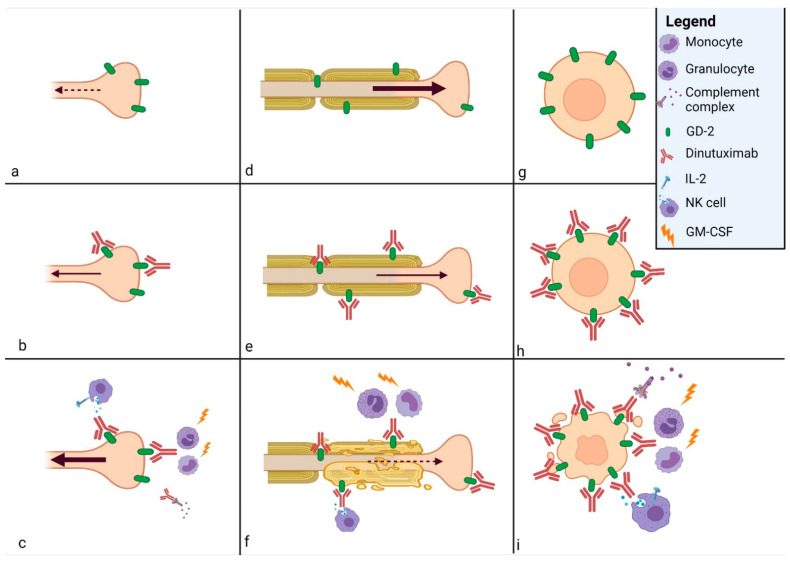
Mechanisms of action of dinutuximab beta. GD2 is expressed on (**a**) nociceptive fiber, (**d**) motor and sensory fibers, and (**g**) neuroblastoma cell membrane. The complex antigen–anti-GD2-antibody is formed on the surface of (**b**) nociceptive fiber, (**e**) motor and sensory fibers, and (**h**) neuroblastoma cell membrane. The complex antigen GD2-antibody activates the antibody-dependent cellular cytotoxicity and the complement-dependent cyto-toxicity, which cause (**c**) neuropathic pain on nociceptive fiber, (**f**) peripheral neuropathy on motor and sensory fibers, and (**i**) apoptosis of neuroblastoma cell. IL-2 acts on Natural Killer (NK) cells and granulocyte-macrophage colony-stimulating factor (GM-CSF) on monocytes and granulocytes to enhance the immunological response.

**Figure 3 ijms-22-12648-f003:**
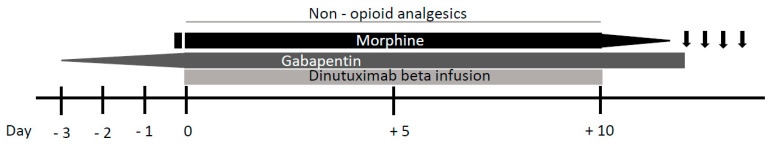
Neuropathic pain management during continuous dinutuximab beta infusion. Gabapentin administration is started three days prior to dinutuximab beta infusion and increased up to 10 mg/kg for three daily administrations. After a bolus, morphine is commenced just before dinutuximab beta infusion and continued as a 24 h intravenous infusion (0.03 mg/kg/h). Following reduction of its infusion rate, morphine is stopped 4 h after the end of dinutuximab beta infusion. If neuropathic pain persist after the intravenous morphine weaning off, oral morphine sulphate or tramadol can be administered on demand.

## Data Availability

No new data were created or analyzed in this study. Data sharing is not applicable to this article.

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
