# Peer review of "Mechanisms, Characteristics, and Treatment of Neuropathic Pain and Peripheral Neuropathy Associated with Dinutuximab in Neuroblastoma Patients"

_ijms, 2021, doi:10.3390/ijms222312648_

Round 1

Reviewer 1 Report

This review provides an interesting overview about side effects and therapeutic implications of ant-GD2 antibodies.

Major concerns:

It would be important to clearly separate the neuropathy from the pain: “neuropathic pain” would be adequate for pain that occurs based on neuropathic neuronal changes – however, while onset of pain is immediate, phasic and is reducing upon repetition, neuropathy has a slow onset and is progressive. Therefore, the acute pain would be best termed as acute nociceptive pain. Notwithstanding, in part of the patients a developing neuropathy might be linked to neuropathic pain, but this is not the main topic of this review.

It is important to note, that there is a puzzling mismatch between the time course of pain and the pharmacokinetics – the authors note, that pain is particularly strong during infusions and gradually declines thereafter. On the other hand, systemic levels of the antibodies have a very long half-life. Even though the mechanism is unclear this aspect should be noted.

Further comments:

Mode of action: the authors should be more specific when it comes to the site of action: “the myelin” should probably read Schwann cell membrane. This is of particular importance as the large antibodies might not easily pass the blood-nerve barrier and for a direct neuronal activity either have to pass the Schwann cells or act directly on the free nerve endings.

Minor comments

„Pain fibers“ should be „nociceptors“

Page 8 second para “Grade >3”  - should be specified (3 of 5, 3 of 10,…)

Author Response

Response to Reviewer 1 Comments

Point 1: It would be important to clearly separate the neuropathy from the pain: “neuropathic pain” would be adequate for pain that occurs based on neuropathic neuronal changes – however, while onset of pain is immediate, phasic and is reducing upon repetition, neuropathy has a slow onset and is progressive. Therefore, the acute pain would be best termed as acute nociceptive pain. Notwithstanding, in part of the patients a developing neuropathy might be linked to neuropathic pain, but this is not the main topic of this review.

Response 1: Thank you so much for the comment. We have modified the article and have distinguished two different paragraphs, one for neuropathy and one for neuropathic pain pathogenesis. We have also modified Figure 2 distinguishing neuropathic pain and neuropathy

Point 2: It is important to note, that there is a puzzling mismatch between the time course of pain and the pharmacokinetics – the authors note, that pain is particularly strong during infusions and gradually declines thereafter. On the other hand, systemic levels of the antibodies have a very long half-life. Even though the mechanism is unclear this aspect should be noted.

Response 2: We agree with the reviewer. On page 2 we have inserted the following sentence “It usually begins within an hour from the start of dinutuximab beta infusion and it is limited to the time of administration of this drug, ending shortly after the termination of the infusion; it usually occurs during the first infusion of the drug and decreases after each course [22].”

Point 3: Mode of action: the authors should be more specific when it comes to the site of action: “the myelin” should probably read Schwann cell membrane. This is of particular importance as the large antibodies might not easily pass the blood-nerve barrier and for a direct neuronal activity either have to pass the Schwann cells or act directly on the free nerve endings.

Response 3: We modified the text and specified the site of action of the antibodies with the following sentence on page 7 “GD2-induced neuropathic pain is mediated by the reactivity of the antibody with the GD2 antigen on the surface of peripheral nerve fibers, particularly C-fibers [31]. The inabil-ity of GD2 antibodies to cross the blood brain barrier implicates the involvement of the pe-ripheral nervous system [31].”

Point 4: Minor comments

„Pain fibers“ should be „nociceptors“

Page 8 second para “Grade >3”  - should be specified (3 of 5, 3 of 10,…)

Response 4: We modified “pain fibers in nociceptors”.

In the new version of the article this sentence is on page 7 paragraph 4; we wrote “grade ≥3 (of a 1 to 5 scale).

Reviewer 2 Report

This review manuscript mainly focusing on the neurotoxicity induced by dinutuximab in neuroblastoma patients. Literature coverage is unsatisfactory, only 30% references are to the publications of the last 5 years. In general, the written of this article is not satisfied, the material is poorly organized. I am opposed to the manuscript being published in the special issue “Mechanisms of Chemotherapy-Induced Peripheral Neuropathy”.

  1. The writing of the Introduction section is unsatisfactory, there is a lack of logical flow and over splitting, the content is redundant, major revision is required.
  2. The title of the manuscript is “Mechanisms, Characteristics, and Treatment of Neuropathic Pain and Peripheral Neuropathy Associated with Dinutuximab in Neuroblastoma Patients.”, while only section 6 and 7 are discussing the neurotoxicity induced by Dinutuximab, which is unacceptable. The discussion of the mechanisms and characteristics of this side effect are lacking, and the treatment of this toxicity is inefficiency.

Author Response

Response to Reviewer 2 Comments

Point 1: This review manuscript mainly focusing on the neurotoxicity induced by dinutuximab in neuroblastoma patients. Literature coverage is unsatisfactory, only 30% references are to the publications of the last 5 years. In general, the written of this article is not satisfied, the material is poorly organized. I am opposed to the manuscript being published in the special issue “Mechanisms of Chemotherapy-Induced Peripheral Neuropathy”.

Response 1: Thank you so much for the comment. We tried to make the most of his advice in order to improve the article.

Point 2: The writing of the Introduction section is unsatisfactory, there is a lack of logical flow and over splitting, the content is redundant, major revision is required..

Response 2: We have reduced the introduction by trying to make it more fluid. We have eliminated paragraphs to avoid repetition in the text.

Point 3: The title of the manuscript is “Mechanisms, Characteristics, and Treatment of Neuropathic Pain and Peripheral Neuropathy Associated with Dinutuximab in Neuroblastoma Patients.”, while only section 6 and 7 are discussing the neurotoxicity induced by Dinutuximab, which is unacceptable. The discussion of the mechanisms and characteristics of this side effect are lacking, and the treatment of this toxicity is inefficiency.

Response 3: We focused more on the pathogenetic mechanisms of dinutuximab neuropathy and neuropathic pain. Now paragraphs 4,5, 6 and sub-paragraphs 4.1 and 5.1 are devoted to the main topic of the article. We have also added subsections 4.1 and 5.1 to further explore the treatment of neuropathy and neuropathic pain.

Round 2

Reviewer 2 Report

The authors have addressed all my concerns and revised accordingly.